A heritability-based comparison of methods used to cluster 16S rRNA gene sequences into operational taxonomic units

Jackson Matthew A. matthew.jackson@kcl.ac.uk
Bell Jordana T.
Spector Tim D.
Steves Claire J.
Department of Twin Research & Genetic Epidemiology, King’s College London, University of London , London , United Kingdom
Chen Jun
Electronic publication date: 2016 Aug 30
Publication date: 2016
Volume: 4
Electronic Location ID: e2341
Received 2016 Jun 7; Accepted 2016 Jul 18
Copyright: ©2016 Jackson et al.
Copyright year: 2016
Copyright holder: Jackson et al.
License: This is an open access article distributed under the terms of the Creative Commons Attribution License, which permits unrestricted use, distribution, reproduction and adaptation in any medium and for any purpose provided that it is properly attributed. For attribution, the original author(s), title, publication source (PeerJ) and either DOI or URL of the article must be cited.
License URL: https://creativecommons.org/licenses/by/4.0/

Keywords: Ecology, Microbiology, Computational biology

Funding: National Institutes of Health (NIH) RO1 DK093595 DP2 OD007444 Wellcome Trust; European Community’s Seventh Framework Programme FP7/2007-2013 National Institute for Health Research (NIHR) Chronic Disease Research Foundation (CDRF) The TwinsUK microbiota project was funded the National Institutes of Health (NIH) RO1 DK093595, DP2 OD007444. TwinsUK received funding from the Wellcome Trust; European Community’s Seventh Framework Programme (FP7/2007-2013), the National Institute for Health Research (NIHR)-funded BioResource, Clinical Research Facility and Biomedical Research Centre based at Guy’s and St Thomas’ NHS Foundation Trust in partnership with King’s College London. CJS is funded under a grant from the Chronic Disease Research Foundation (CDRF). The funders had no role in study design, data collection and analysis, decision to publish, or preparation of the manuscript.

==============================
A variety of methods are available to collapse 16S rRNA gene sequencing reads to the operational taxonomic units (OTUs) used in microbiome analyses. A number of studies have aimed to compare the quality of the resulting OTUs. However, in the absence of a standard method to define and enumerate the different taxa within a microbial community, existing comparisons have been unable to compare the ability of clustering methods to generate units that accurately represent functional taxonomic segregation. We have previously demonstrated heritability of the microbiome and we propose this as a measure of each methods’ ability to generate OTUs representing biologically relevant units. Our approach assumes that OTUs that best represent the functional units interacting with the hosts’ properties will produce the highest heritability estimates. Using 1,750 unselected individuals from the TwinsUK cohort, we compared 11 approaches to OTU clustering in heritability analyses. We find that de novo clustering methods produce more heritable OTUs than reference based approaches, with VSEARCH and SUMACLUST performing well. We also show that differences resulting from each clustering method are minimal once reads are collapsed by taxonomic assignment, although sample diversity estimates are clearly influenced by OTU clustering approach. These results should help the selection of sequence clustering methods in future microbiome studies, particularly for studies of human host-microbiome interactions.

Introduction

The field of microbiome research has seen rapid expansion this last decade (Jones, 2013). One of the techniques most frequently used to profile microbial communities is 16S rRNA gene sequencing, where PCR amplification of variable marker regions is used to determine a sample’s microbial composition (Pace, 1997). The taxonomic resolution of sequence variation across a marker region is limited both biologically and technically, because sequence divergence may not represent wider biological divergence between taxa (Stackebrandt & Goebel, 1994; Mignard & Flandrois, 2006), and sequencing errors introduce artificial divergence (Huse et al., 2010; Schloss, Gevers & Westcott, 2011). As a result, it is not necessarily useful to enumerate every unique sequence observed particularly given that samples may contain hundreds of thousands of unique reads. To simplify analyses, reads within a 16S rRNA gene dataset are typically collapsed into operational taxonomic units (OTUs). This is carried out based on sequence similarity between reads. Convention is typically to group reads that share at least 97% identity, which is considered “species” level. Although collapsing can be carried out to any threshold and there is no clear definition of what constitutes a bacterial species.

A variety of methods are available to collapse 16S data to OTUs (Edgar, 2010; Edgar, 2013; Rognes et al., 2016; Mercier et al., 2013; Mahé et al., 2014; Schloss & Handelsman, 2005; Eren et al., 2014), often implemented within software wrappers such as QIIME and Mothur (Caporaso et al., 2010; Schloss et al., 2009). One of the main divides in approaches is whether experimental sequences are clustered against a reference database of sequences (Liu et al., 2008), termed closed reference clustering (Navas-Molina et al., 2013), or solely clustered within the experimental data itself, generating what are termed de novo OTUs (Schloss & Handelsman, 2005; Navas-Molina et al., 2013). Closed reference clustering is computationally more efficient given that each sequence should maximally only be compared against each reference sequence, whereas de novo clustering could require pair-wise comparisons between all experimental reads. Closed reference approaches also facilitate comparisons between datasets as OTUs can be defined and matched based on their reference sequences; however, reads which do not match any reference sequences will be discarded. De novo clustering does not have this limitation and includes all experimental reads in resultant OTUs, which may better represent rare and novel taxa (Navas-Molina et al., 2013). A third approach, termed open-reference clustering, aims to capitalise on the benefits of both approaches by first clustering experimental sequences against a reference followed by de novo clustering of discarded sequences (Navas-Molina et al., 2013).

Once a reference or de novo based approach has been selected, a number of different algorithms can be used to cluster sequences by similarity (Schloss & Handelsman, 2005; Caporaso et al., 2010; Edgar, 2010; Edgar, 2013; Rognes et al., 2016; Mercier et al., 2013; Mahé et al., 2014; Eren et al., 2014). Linkage based methods calculate pairwise distances between all sequences allowing hierarchical clustering to OTUs (Schloss & Handelsman, 2005). There are also multiple greedy algorithms available, which aim to reduce computation time using heuristic approaches to finding optimal groups without calculating all possible distances (Edgar, 2010; Edgar, 2013; Rognes et al., 2016). Furthermore, there have been a number of methods proposed to summarise 16S data without using a predetermined global similarity threshold. These include simply using de-replicated sequences (reads collapsed by 100% similarity), defining OTUs by inherent separation within the dataset using local rather than global cut-offs (Mahé et al., 2014), and splitting reads into groups based on sequence entropy at each position in aligned reads (Eren et al., 2014).

With the range of available approaches to OTU picking some comparative metric is required to assess their performance. Previously, clustering algorithms have been compared based on a number of metrics including: their computational efficiency (Edgar, 2010; Kopylova et al., 2016; Chen et al., 2013); the number of OTUs they produce (Schmidt, Rodrigues & Von Mering, 2015; Kopylova et al., 2016; Chen et al., 2013); the accuracy of the similarity between sequences within their OTUs (Westcott & Schloss, 2015; Schloss, Gevers & Westcott, 2011; Schloss, 2016); their ability to handle sequencing artefacts (Edgar, 2013); their reconstruction of simulated data sets (Kopylova et al., 2016; Chen et al., 2013); the similarity between method outputs (Schmidt, Rodrigues & Von Mering, 2015; Kopylova et al., 2016); and the reproducibility of their clustering within subsets of the same data (He et al., 2015). However, the optimal approach between de novo and reference clustering, and the different clustering algorithms is dependent on which measure of quality is considered.

As there is no accepted standard for definition and enumeration of microbial taxa in a community, existing comparison metrics have exclusively dealt with technical aspects of clustering. It is not clear which of these metrics is most important in determining a methods ability to generate OTUs most representative of the biological units underlying microbial community structure. Here we suggest heritability as a measure of the biological relevance of OTUs.

Heritability quantifies the percentage of phenotypic variation that is attributable to genetic variability. Twin studies are a well-established method for estimating heritability. These compare the correlation of phenotypes within monozygotic (MZ) twin pairs whom share identical nuclear DNA, to the correlations within dizygotic (DZ) pairs whom on average share half their genetic material. Variation in a phenotype can then be apportioned into variation due to genetic factors, which are shared by twins to a varying degree, based on zygosity and to environmental factors, which are not shared by twins (Franic et al., 2012; Boomsma, Busjahn & Peltonen, 2002).

TwinsUK is a long established cohort of unselected British twins (Moayyeri et al., 2013). 16S rRNA gene sequencing of faecal samples from the cohort has been used to demonstrate heritability of the microbiome (Goodrich et al., 2014; Goodrich et al., 2016), and to identify a number of phenotype-microbiome associations (Jackson et al., 2016a; Jackson et al., 2016b; Barrios et al., 2015). Under the assumption that some heritability within the microbiome is acting at the level of individual taxa-host interactions, we propose that the heritability of OTUs is representative of their ability to summarise the underlying biological units within a microbial community.

Here we compare heritability estimates of 11 different methods of summarising 16S reads from 1,750 faecal samples of 473 MZ and 402 DZ twin pairs. Overall, we find that de novo clustering, regardless of algorithm, consistently produces more heritable OTUs than reference based approaches, with VSEARCH and SUMACLUST producing the highest heritability estimates from those considered. No difference in heritability was observed once OTUs had been collapsed by taxonomic assignment. We also find that clustering method can influence relative sample diversity, dependant on the diversity metric used. These results should provide guidance to researchers in selecting the appropriate approach to OTU picking, in particular in studies investigating human host-microbiome interactions.

Methods

Faecal sampling and 16S rRNA gene sequencing

Analyses were carried out using 16S rRNA gene sequencing reads from a subset of published data from the TwinsUK cohort. Sample collection, DNA extraction and sequencing have previously been reported (Goodrich et al., 2014). In brief, twins produced the sample at home, which was then kept refrigerated and/or on ice before freezing at −80 °C in the TwinsUK laboratory at King’s College London. Frozen samples were then shipped to Cornell University where extracted DNA from samples was PCR amplified over the V4 variable region of the 16S gene. The resulting amplicons were multiplexed and sequenced using the Illumina MiSeq platform to generate 250 bp paired-end reads. Ethical approval for microbiota studies within TwinsUK were provided by the NRES Committee London—Westminster (REC Reference No.: EC04/015). All participants provided written consent.

Pre-processing of sequencing reads

Paired reads were joined using fastqjoin, within QIIME (Caporaso et al., 2010), discarding reads without a minimum overlap of 200 nt and those containing ambiguous bases. Joined reads were de-multiplexed also removing barcodes. The data were filtered to only include the subset of 1,750 samples from the 473 MZ and 402 DZ complete twin pairs used in these analyses. Within this set, there were 158,635,772 reads with an average of 91,170 reads per sample. These were split per sample and de novo chimera checking carried out on each individually using USEARCH de novo chimera detection in QIIME with a no vote weight of 7 (Edgar et al., 2011; He et al., 2015). This identified an average of 8,471 chimeric reads per sample all of which were removed. Sample reads were then concatenated to one file and all sequences <252 nt or >253 nt in length discarded (<1% of reads) (Kozich et al., 2013). After chimera removal and length filtering, the final data set contained 142,307,280 reads across all samples. This fasta file was used as the input for all 16S collapsing approaches.

These reads and associated metadata, covering a larger selection of samples and twins than the subset described here, are available from the European Nucleotide Archive (ENA) from the study with accession number ERP015317 (Goodrich et al., 2016).

Clustering of 16S rRNA gene sequencing reads

All threshold based OTU clustering approaches and Swarm were implemented using QIIME 1.9.0 (Caporaso et al., 2010; Mahé et al., 2014). VSEARCH de novo clustering was implemented within the QIIME wrappers using an alias to run VSERARCH in place of USEARCH (Rognes et al., 2016; Edgar, 2010). VSEARCH is not restricted to the same memory limitations as the free version of USEARCH, enabling its use across our whole data set. It also accepts the same commands for de novo clustering so required no alterations to the QIIME wrapper. Where a reference was required, the Greengenes reference and taxonomy version 13_8 was used (DeSantis et al., 2006). De-replicated sequences were generated using VSEARCH (Caporaso, 2015). Minimum entropy decomposition (MED) was run from scripts within the oligotyping pipeline using default parameters (Eren et al., 2014; Eren et al., 2013). An overview of how each clustering method works, the clustering pipeline, and complete commands used for each clustering procedure can be found in Supplemental Information 1.

Heritability analyses

Heritability of microbiome traits was calculated in a manner similar to as previously reported (Goodrich et al., 2014). Estimates were calculated for OTUs found in at least 50% of samples as OTU absence, which skews the distribution of abundances, would be less influential on model fitting. A pseudo count of 1 was added to all OTUs to remove absent data in the resultant OTU tables of each clustering approach. Counts were converted to within sample relative abundances and tables subset to only include OTUs found in at least 50% of samples (prior to the addition of pseudo counts). The powerTransform package in R was used to estimate a Box–Cox transform lambda producing approximately normally distributed residuals from a linear model with OTU abundance as a response and gender, age, sequencing run, sequencing depth, how the sample was collected, and the technician who loaded and extracted the DNA as predictors. This was carried out for each OTU and the transformed residuals used in heritability estimation.

Estimates were found by fitting OTU abundances to a twin-based ACE model. This estimates narrow-sense heritability (the heritability due to additive genetic effects—A) on the assumption that variance resulting from shared environment (common environment—C) is equal in MZ and DZ twins, with remaining variance attributed to environmental influences unique to individuals (E) (Franic et al., 2012). Maximum likelihood estimates were found by structural equation modelling using OpenMX in R (R Development Core Team, 2009; Boker et al., 2011). Heritability estimates for collapsed taxonomic traits were calculated in the same manner as for OTUs.

Between method comparisons of OTU heritability and other distributions were carried out in R using pairwise Mann–Whitney U tests using Benjamini–Hochberg FDR correction to account for multiple testing.

Alpha diversity calculation and taxonomic assignment

Each complete OTU table was rarefied to 10,000 sequences 25 times. Alpha diversity calculation was carried out on each rarefied table for each method using Simpson, Shannon, Chao1 and raw OTU count metrics, with final diversity values taken as the mean across all rarefactions. Alpha diversity estimates were compared using Mann–Whitney U tests to contrast absolute values between methods and Kendall rank correlations to compare sample rankings between methods.

For each clustering method, except closed reference, representative sequences were selected as the most abundant read within each OTU. These were then used to assign taxonomy against the Greengenes 13_8 database with a 97% similarity threshold using the UCLUST method in the assign taxonomy script of QIIME. OTU tables were collapsed based on taxonomic assignment at all levels from genus to phylum. Differences in heritability of taxa between methods were compared using a generalised linear model in R, to determine the ability of taxonomic assignment and clustering method to predict heritability estimates as the response variable. This was carried out across all taxonomic levels considering all taxa that were found across all 11 clustering approaches.

Results

De novo clustering produces more heritable OTUs than closed reference clustering

16S microbiome profiles were available for 473 MZ and 402 DZ pairs within previously reported data. Joined paired end read data were revisited and chimeric sequences removed on a per sample basis. Total read data across all 1,750 samples was then clustered using de novo, closed reference, and open reference approaches using the UCLUST algorithm (Edgar, 2010), the current default in QIIME, to form OTUs with a threshold similarity of 97%. The resultant OTU tables are summarised in Table S1. De novo clustering produced more OTUs than closed reference and as a result, a more sparsely distributed OTU table. Open reference picking was an intermediate of the two approaches as might be expected.

Figure 1 Twin based A, C, and E estimate comparisons between closed and open reference, and de novo clustering using UCLUST with a similarity threshold of 97%.

(A) Boxplots representing the A, C and E estimates for all OTUs found in at least 50% of samples in each method. De novo clustering A estimates significantly higher than those of closed reference clustering (q = 0.017). (B) The same estimates as in A but displayed as a density function showing the distribution of estimates amongst OTUs.

Across all three methods the A, C, and E estimates were within the range expected from previous reports within the cohort (Goodrich et al., 2014; Goodrich et al., 2016). De novo clustering produced OTUs with significantly higher (q = 0.017) heritability (A) estimates than closed reference clustering (Fig. 1A). De novo heritability estimates were also higher than those of open reference OTUs although the difference was non-significant. There were no significant differences in the distributions of C estimates between any methods. De novo clustering produced OTUs with significantly lower E estimates than both closed (q = 0.02) and open reference (q = 0.003) approaches.

Whilst significant, the difference in OTU heritability estimates was only moderate. The mean of the de novo A estimates was 1% higher than that of the closed reference clustered OTUs. However, the distribution of A, C, and E estimates were also divergent, as shown in Fig. 1B. Closed reference A estimates displayed a bimodal distribution with OTUs either having no or little heritability with fewer highly heritable units. De novo clustering produced units of higher heritability whose estimates were more evenly distributed. Open reference clustering displayed features of both distributions resulting in higher levels of moderately heritable OTUs.

VSEARCH and SUMACLUST produce more heritable de novo OTUs than UCLUST

As de novo clustering produced the most heritable OTUs using UCLUST, we aimed to determine the influence of using alternative threshold based algorithms for clustering. Linkage based clustering approaches were not considered as it was unfeasible to generate distance matrices between the large number of unique reads within the data set. OTUs were clustered at 97% similarity using two alternate greedy algorithms within QIIME—VSEARCH and SUMACLUST (Rognes et al., 2016; Mercier et al., 2013). The open-source algorithm VSEARCH was used in place of the QIIME default USEARCH to overcome the memory limitations of its free version. VSEARCH has previously been shown to match or outperform USEARCH in terms of accuracy (Westcott & Schloss, 2015). Clustering with VSEARCH was carried out using both distance and abundance options as tiebreak assignments. The resultant OTU tables are summarised in Table S1.

There were no significant differences in the mean magnitudes of the A, C, or E estimates between all four methods tested (Fig. 2A). The distributions of estimates were very similar in the SUMACLUST, and both VSEARCH approaches (Fig. 2B). UCLUST OTUs contained a higher proportion of A estimates falling between 0.05 and 0.15, with the other methods containing higher proportions of more heritable OTUs. The VSEARCH methods had more OTUs with high heritability estimates (0.35–0.4), with the distance tiebreaker based method producing slightly fewer. SUMACLUST produced the most heritable OTU. Overall, all de novo algorithms produced estimates higher than the UCLUST reference based approaches at a threshold of 97% similarity, with SUMACLUST and VSEARCH approaches producing more heritable OTUs than UCLUST.

Figure 2 Twin based A, C, and E estimate comparisons between different greedy algorithms for de novo clustering at a 97% similarity threshold.

(A) Boxplots representing the A, C and E estimates for all OTUs found in at least 50% of samples in each method. There was no significant difference in A estimates between methods. (B) The same estimates as in A but displayed as a density function showing the distribution of estimates amongst OTUs.

Clustering at higher thresholds and other alternatives to clustering

We aimed to investigate the use of more stringent thresholds repeating VSEARCH abundance based clustering with identity thresholds of 98 and 99%, and simply de-replicating the sequences, the equivalent of a 100% threshold. We also clustered sequences using two approaches that do not rely on a sequence identity threshold—MED and Swarm (described in Supplemental Information 1) (Eren et al., 2014; Mahé et al., 2014). Of the thresholds, 97% produced the most heritable OTUs (Fig. 3A), whose distribution of A estimates was significantly different to those of the 99 (q = 0.02) and 100% (q = 0.0001) cut-off OTUs (Fig. 3B). As the percentage identity increased from 97% through to 100% the distribution of A estimates became less continuous, with small groups of units with high heritability and much larger numbers with low heritability. This suggests that in some instances, the heritability estimate of an OTU clustered at 97% identity may be driven by an individual, highly heritable sequence; as opposed to the accumulative effects of the variance across all its reads.

Figure 3 Twin based A, C, and E estimate comparisons between three different thresholds of de novo clustering using VSEARCH, VSEARCH de-replicated sequences, and two non-threshold based techniques.

(A) Boxplots representing the A, C and E estimates for all OTUs found in at least 50% of samples in each method. The 97% threshold produced significantly more higher A estimates than the 99 and 100% thresholds (q = 0.02, q = 0.0001). (B) The same estimates as in A but displayed as a density function showing the distribution of estimates amongst OTUs.

MED produced very few units in total (Table S1). However given this broad level of summary, which is comparable to that of closed reference clustering, the resultant units A estimates were not significantly different to VSEARCH OTUs clustered at the 97% level. Similarly, the heritability of OTUs resulting from clustering by Swarm had heritability’s within the range of the VSEARCH methods, however the distribution of A estimates more closely resembled OTU clustering at a threshold of 99%.

De novo clustering at 97% generates more heritable OTUs than reference-based approaches when considering only heritable units

The power of a twin study to detect and accurately estimate the additive genetic variance of a trait is limited by the total number of pairs and the proportion of MZ twins considered (Visscher, 2004). As noise in the A estimates for non and low heritability traits may influence the overall distribution, we compared A estimate distributions across all previously clustered techniques considering only heritable OTUs—those with A estimates greater than the mean of all OTUs (8%) and with a lower 95% confidence interval of at least 1% (Fig. 4). When only considering the most heritable OTUs, the majority of de novo based approaches produced units with higher heritability estimates than the reference-based approaches. VSEARCH AGC clustering at 97 and 98%, and DGC clustering at 97% produced significantly higher estimates than closed reference UCLUST. As did SUMACLUST de novo clustering (97% identity), which also produced units with significantly higher heritability than those produced by open reference based clustering. De novo clustering at higher sequence identity thresholds (99 and 100%) produced OTUs with significantly lower estimates than SUMACLUST at 97%.

Figure 4 Comparison of A heritability estimates between all clustering approaches. Only considering OTUs who’s A estimate was greater than the mean (∼8%) and had a lower 95% CI greater than 1%.

SUMACLUST and VSEARCH clustering produced OTUs with significantly higher heritability estimates than OTUs produced using reference-based clustering. Significant differences are shown where * indicates q < 0.05 and ** indicates q < 0.01.

Differences resulting from clustering approach are not apparent after collapsing by taxonomic assignment

The ability of a technique to generate OTUs representing fine scale biological units may be less important for studies aiming to identify effects at higher taxonomic levels. To determine if choice of OTU clustering approach significantly effected the ability to generate representative taxa we collapsed each OTU table at all taxonomic levels from genus to phylum, and estimated the heritability of taxa at each level (Table S2). We then investigated the ability of taxonomic assignments and clustering methods to predict taxa heritability estimates. We found that assignments to 150 of the 168 taxa found across all 11 methods were significant predictors of heritability, however none of the clustering methods had a significant effect. This suggests that from genus through to higher-level taxonomic summaries there is sufficient collapsing of reads that the previously observed differences in OTU clustering are not apparent.

Alpha diversity measures are influenced by clustering approach

As the largest difference observed between methods was the number of OTUs generated, we aimed to determine the influence of clustering approach on alpha diversity estimates. The absolute values of sample diversity estimates were significantly different between almost all methods of clustering for all four diversity estimates considered (Fig. 5). In particular, the values of OTU count and Chao1 (richness measures influenced by rarer OTUs) were much higher in the de-replicated (or 100% identity) sequences. These results show that absolute diversity levels are not comparable between methods over the same rarefied data.

Figure 5 Comparison of absolute alpha diversity values for Shannon, Simpson, Chao1, and OTU count indices across all samples.

OTU tables for each method were rarefied to 10,000 sequences 25 times and the mean diversity calculated across all tables. There was a significant difference in the distribution of diversity values between all methods for all four metrics. De-replicated sequences in particular inflate richness-based measures.

To determine if these differences would influence comparative diversity analyses, we measured the rank based correlation between methods for each diversity metric (Fig. 6). For both the Shannon and Simpson metrics the diversity rankings were highly correlated (τ > 0.6, mean = 0.83) between all methods. However, when using the Chao1 and OTU count metrics there was a reduced correlation between diversity rankings. In particular, the closed reference and MED approaches were poorly correlated with de novo based approaches. This is likely due to under representation of rare sequences as both of these methods discard reads. Our results show that clustering approach can influence the relative diversities between samples in a study dependant on the diversity measure used. This may be particularly important in the interpretation of diversity association analyses, where use of a closed reference approach could produce different results to the use of de novo clustering.

Figure 6 Kendall’s Tau rank based correlations between samples across methods for each of Shannon, Simpson, Chao1 and OTU count metrics.

Rank correlation represents the concordance between relative diversity assignments between the same samples in each clustering method. There is generally high correlation between all methods when using the Shannon and Simpson indices, which measure evenness of species distribution. However, the de-replicated, closed reference, and MED clustered OTUs show poor correlation in the richness measures (Chao1 and OTU count). Clustering method may therefore influence diversity association analyses.

Discussion

Here we propose and demonstrate the use of heritability estimates as a novel approach to methodological comparisons. There is an established taxa dependent variability in the heritability of the gut microbiome (Goodrich et al., 2014). Heritability estimates aim to quantify the percentage of a trait’s variation that is due to the influence of host genetics. Given that bacteria within the microbiome contain a range of functional properties, determined by their own genetics, we assume that the heritability of an OTU is driven by a specific bacteria-host interaction. By this logic, we would expect the OTU clustering approach that best groups reads sourced from bacterial units with similar functional properties to produce OTUs with the highest heritability estimates.

Using the distribution of heritability estimates as a measure of biological representation, we have demonstrated that de novo clustering produces OTUs that are more representative of functional microbial units than reference based approaches. We have also shown that within the various algorithms considered VSEARCH and SUMACLUST produced the most representative OTUs. Within our comparison of clustering thresholds, we found that 97% sequence identity produced the most heritable units when compared to more stringent cut-offs. We have shown that these effects are only applicable at the OTU level, as clustering approach does not significantly influence the heritability estimates of collapsed taxonomies. Finally, we have demonstrated that choice of clustering approach can effect both absolute and relative diversity measures with implications for comparisons across microbial studies.

The aim of OTU clustering is to group sequences based on sequence similarity. Our comparisons are based on the assumption that the genetic relatedness between 16S reads is related to the functional similarity between their bacterial sources. In this way, a clustering method that best groups reads with similar sequence will also groups reads from bacteria with similar functional relationships to the host. These methods should therefore produce the highest heritability estimates, as they will produce less noise in the variance of OTU abundances due to incorrectly grouped read counts. Whilst this may not provide an accurate quantification of the quality of sequence identity within OTUs (as provided by existing methods discussed below), it does provide a measure of the functional representation of the units. For example, in our data the OTUs clustered with 99 and 100% identity thresholds produced lower heritability estimates. Suggesting that 97% is the best threshold to generate units that represent functional units within the microbiome. A methods ability to represent functional units is arguably of more importance than genetic accuracy, particularly for studies in areas such as human microbiome research where the goal is often to identify the functional roles of microbes in human health.

Recently, four studies were published that each compared multiple OTU clustering approaches (He et al., 2015; Kopylova et al., 2016; Westcott & Schloss, 2015; Schloss, 2016). The first used the stability of sequence assignments within subsets of the same data sets as a measure of quality, finding that reference based approaches outperformed de novo clustering (He et al., 2015). The heritability comparisons presented here do not reflect these findings, suggesting that stability does not relate to functional representation. However, stability may be an important consideration for studies comparing across data sets. Our findings also suggest that reference based approaches would be sufficient when analyses are only concerned with collapsed taxonomies.

Two studies have compared clustering methods using Matthew’s correlation coefficient (MCC) to quantify their accuracy in clustering sequences sharing 97% sequence identity (Westcott & Schloss, 2015; Schloss, 2016). They found that de novo clustering produced more accurate OTUs than reference based approaches (Westcott & Schloss, 2015), and that VSEARCH and SUMACLUST out performed Swarm in terms of OTU accuracy (Schloss, 2016). The differences between reference and de novo OTUs in our heritability estimates, whilst moderate, were significant and broadly agreed with these observations. This suggests that accuracy is also representative of the biological representation of OTUs. This might be expected under the assumption that sequence similarity, at least in part, reflects functional similarity.

Kopylova et al. (2016) compared a number of clustering methods using a variety of measures from recapitulation of simulated data to inter-method correlations. Within the methods considered here, they found that Swarm, SUMACLUST and UCLUST, performed equally well at reconstructing expected taxonomies from simulated data but differed in the number of OTUs produced and subsequently produced different absolute diversities, a finding also described by Schmidt, Rodrigues & Von Mering (2015). Differences in absolute measures would be expected given the variation in OTU numbers between methods. We have also shown that these differences can influence the relative diversity rankings between samples and suggest caution in the interpretation of comparative diversity analyses when using closed reference clustering and community richness metrics.

Overall, across previous comparisons of greedy clustering algorithms in combination with the heritability results we have presented here, VSEARCH and SUMACLUST seem to produce the best combination of accuracy, stability and heritability. We would therefore recommend either of these approaches for de novo clustering. SUMACLUST and USEARCH are currently available within QIIME. VSEARCH has recently been implemented within Mothur (Westcott, 2016), and QIIME 2 will integrate VSEARCH for OTU clustering and de-replication (Greg Caporaso, personal communication, 15th April 2016). Based on our threshold comparisons a similarity cut-off of 97% appears optimal, however this threshold may be specific to VSEARCH application to faecal samples as optimal thresholds can vary by the complexity of the microbial communities under investigation and the method used (Chen et al., 2013).

Whilst we tried to include the most frequently used approaches, our study is not comprehensive. We restricted the majority of our comparisons to clustering algorithms that were available within the QIIME pipeline; however, even in this respect, our comparison was not exhaustive. There are further reference based clustering algorithms such as BLAST and SortMeRNA that were not considered (Camacho et al., 2009; Kopylova, Noe & Touzet, 2012), and de novo approaches such as USEARCH and CD-HIT (Edgar, 2010; Li & Godzik, 2006). We chose to implement clustering via QIIME as it is one of the most widely used methods to generate OTUs and provided stability in other areas of the processing pipeline, such as taxonomic assignment, which improved comparability. However, QIIME does not implement all OTU clustering algorithms and all of those compared here can also be run independently of QIIME, with a number of them having newer versions available that could influence clustering. Our comparison is also limited by the exclusion of linkage-based approaches, as typically implemented using the Mothur pipeline (Schloss et al., 2009). These were not considered in our comparison due to the high computational burden of generating the pair-wise sequence distance matrices that these methods require. Computing time and memory limits were met even when applying additional sequence filtering or restricting distance calculation by taxonomy (Kozich et al., 2013). Previous MCC accuracy comparisons showed that average based linkage clustering were as or more accurate than the best de novo approaches dependent on the dataset considered (Schloss, 2016). Given the reflection between the MCC and heritability results we might speculate that average linkage based approaches could produce biologically relevant units equivalent to the de novo algorithms we considered.

Our comparisons are further limited as we have only considered sequencing from human faecal samples of a single population. A sufficiently large sample is required to determine heritability estimates for moderately heritable traits (Martin et al., 1978); however, clustering and analysis of data on this scale is time consuming and computationally intensive, making it non-trivial to incorporate additional data. There are also few twin microbiome data sets available at the scale of TwinsUK. It is known that existing measures of clustering quality can be data set dependent (Schloss, 2016; Chen et al., 2013; Kopylova et al., 2016). Therefore, our results may not be applicable to non-faecal samples. However, they should be of particular relevance when experiments aim to study the functional aspects of the human gut microbiome.

In conclusion, heritability analyses can be used to provide a measure of the quality of the functional representation of OTUs. This may be used for additional guidance in selecting an appropriate clustering approach in combination with the other comparative metrics available, although the optimum method will be largely dependent on each studies experimental and analytical requirements.

Supplemental Information

Supplemental Information 1 Supplementary Methods

Details of the analysis pipeline used and an outline of each clustering method.

Click here for additional data file.

Table S1 Summary of OTU counts resulting from each method

Click here for additional data file.

Table S2 Method-wise A,C and E estimates for each taxa found across all methods

Click here for additional data file.

We would like to thank Julia Goodrich, Andrew Clark, and Ruth Ley of the Department of Molecular Biology and Genetics at Cornell University, our collaborators on the collection, processing, and analysis of the TwinsUK 16S gut microbiome data, whom provided guidance and comments on this manuscript.

Additional Information and Declarations

Competing Interests

Author Contributions

Human Ethics

Data Availability

The authors declare there are no competing interests.

Matthew A. Jackson conceived and designed the experiments, performed the experiments, analyzed the data, wrote the paper, prepared figures and/or tables, reviewed drafts of the paper.

Jordana T. Bell and Tim D. Spector contributed reagents/materials/analysis tools, reviewed drafts of the paper.

Claire J. Steves contributed reagents/materials/analysis tools, reviewed drafts of the paper, supervised all work carried out by Matthew A. Jackson.

The following information was supplied relating to ethical approvals (i.e., approving body and any reference numbers):

Ethical approval for microbiota studies within TwinsUK were provided by the NRES Committee London—Westminster (REC Reference No.: EC04/015).

The following information was supplied regarding data availability:

Sequencing data used within these experiments is available as part of the European Nucleotide Archive (ENA) study with the accession number ERP015317.

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
