# Peer review of "A heritability-based comparison of methods used to cluster 16S rRNA gene sequences into operational taxonomic units"

_PeerJ, doi:10.7717/peerj.2341_

## Round 0.1 · original submission · Minor Revisions

The manuscript presents an interesting comparison of different OTU-picking methods using the 'heritability' criterion. The study could be further strengthened by including some other popular methods and using the most recent version of UCLUST as one reviewer noted. The ROC analysis suggested by one reviewer may not be necessarily performed since the Wilcoxon rank sum test P values should reflect the AUCs. The community will also benefit from being provided with the link to the FASTA files, mapping file and scripts for reproducing their results.

One more point:
The author stated that "When only considering the most heritable OTUs de novo based approaches all produced more heritable units than the UCLUST reference based approaches". However, Supplementary Table 2 shows the opposite with UCLUST closed reference method produced the most heritable units (n=114). Is it a typo?

Reviewer 1 ·

Basic reporting

This paper is well written. However, there are some long sentences difficult to understand. For example “This suggests that from genus through to higher-level taxonomic summaries there is sufficient collapsing of reads that previously observed differences in OTU clustering are not apparent, with all methods generating taxa with similar heritability”. Those long sentences can be split into shorter ones for clarity.

Experimental design

The research questions and data analysis are well designed. The authors provided detailed methods in the supplementary materials.

Validity of the findings

The bioinformatics and statistical analysis are logically sound.

Additional comments

The authors proposed a novel approach to evaluate the OTU clustering methods by using heritability analysis. The idea is quite interesting. The manuscript is well-written and the analysis are logically sound. I have only one minor comment.

As reported in Goodrich et al. 2014, not all OTUs were influenced by host genetics ( 36% OTUs have estimated C > A). Goodrich et al. defined the heritable OTUs as A > 0.2. However, the authors in this manuscript compared methods based on all OTUs and only found moderate difference in some of the analysis. Maybe the authors can refine the analysis to heritable OTUs and non-heritable OTUs with certain A value as cutoff, and use ROC curves to compare different clustering methods.

·

Basic reporting

I found the manuscript clear and well written. Figures are appropriate.

I would appreciate a statement about the availability of the sequencing data used in the study.

The titles on the X-axis on figure 1B, 2B and 3B should be “Heritability estimate”, not just “Estimate”.

The cited paper by Kopylova et al. was published in 2016, not 2014.

The URL to the paper by Boker et al (2011) is incorrect.

There is a typo in the reference to Schmidt et al. (2014): “n/a, n/a”

Experimental design

I would welcome more details, e.g. scripts, about the procedure for calculating the heritability estimates. The rest is described in sufficient detail.

The authors argue that they have excluded linkage based clustering methods (i.e. mothur) due to high computational demands. There are also other popular methods that could have been included, like UPARSE, that should not be more demanding.

More recent versions of UCLUST / USEARCH and Swarm could have been tested. What is the reason for not including these?

I think UCLUST (or USEARCH version 6 or 7) could have been run in AGC mode as well.

Validity of the findings

In this manuscript, the authors compare the ability of various clustering algorithms, tools and parameters to cluster sequences into OTUs. Their approach is based on measuring the heritability of OTUs based on fecal samples from monozygotic and dizygotic twins. To my knowledge, this is the first attempt at using this kind of data to compare OTU clustering methods.

The authors find that there are significant differences between the methods at the OTU level, especially when only the most heritable OTUs were considered. The conclusion is that de novo clustering methods performs better than closed reference clustering. When OTUs were collapsed at higher taxonomic levels the differences disappeared. This indicates that it is the fine grained grouping of organisms in the OTUs that makes a difference.

Perhaps the most debatable issue with this study is how well a higher estimate of heritability of OTUs really reflects more biologically relevant OTUs, and whether this is an appropriate way of comparing methods. I find the approach interesting and intuitively appealing, but I am not convinced how well it really works. I think the approach measures phenotypic properties of the OTUs, while clustering is supposed to cluster the sequences based on genetic properties. Also, would a higher level of heritability necessarily correspond to a more correct clustering? More discussion of these issues would be valuable.

It might be a problem that the set of species that interact with the genetically variable properties of the host is small. The comparison would then depend on just a small set of OTUs and would be more vulnerable to random effects. Based on table S2, it seems like the number of OTUs with high heritability (lower 95% CI of A above 0.1) is between 29 and 47 for the de novo methods. Is this enough?

Line 284-286: “This suggests that in some cases OTU heritability in the less stringent thresholds may be driven by a small number of more heritable sequences, as opposed to the accumulation of multiple smaller effects.“ Isn’t it the exact opposite? With less stringent I would think of 97% similar sequences, while more stringent would be 100%.

Additional comments

Disclosure: I am the main developer of the VSEARCH and Swarm tools included in the comparison in this study. I am also a co-author on the Kopylova et al. (2016) paper.

---

## Round 0.2 · accepted · Accept

The authors have addressed the major concerns adequately. It is suitable for publication now.